# Measurement of inferior colliculus volume based on MRI image stacks and its relationship with age and hearing status

**Pingling Kwok**‡*, **Otto Gleich**‡, **Peter Koch, Gudrun Schenkl, Nina Koch, Christopher Bohr**

Department of Otolaryngology, University Hospital Regensburg, Regensburg, Germany

‡ PK and OG are joint senior authors on this work.
* pingling.kwok@ukr.de

**Data Availability Statement:** All relevant data are contained within the paper and its Supporting information files.

## Abstract

The inferior colliculus is a key nucleus in the central auditory pathway, integrating acoustic stimuli from both cochleae and playing a crucial role in sound localization. It undergoes functional and structural development in childhood and experiences age-related degeneration later in life, contributing to the progression of age-related hearing loss. This study aims at finding out, whether the volume of the human inferior colliculus can be determined by analysis of routinely performed MRIs and whether there is any age-related variation. A further goal is to detect correlations between volume and existing hearing loss of the patients. A retrospective search in the data of the Regensburg ENT department was done. 123 MRI datasets were used to mark the voxels of the inferior colliculus on the MRI layers. The volumes could then be calculated by using the respective DICOM data and were correlated with age, gender and hearing status of the patients. Results suggested that a voxel-based method on routine clinical MRI stacks to determine the volume of the inferior colliculus is possible. The volume shows an age-dependency. There is a growth from infancy until adulthood and a significant decrease in patients over the age of 60 years. Left and right inferior colliculi do not show any systematic asymmetry in volume. There is no difference between females and males. In the group with asymmetric hearing (n = 13) a significant reduction of the volume on the deprived side (p = 0.036) was found. The proportion of subjects with severe hearing loss at least on one side was significantly higher in the old (>60 years) as compared to younger adults (10 to 60 years), suggesting that severe hearing loss may be associated with a reduced volume of the inferior colliculus in aged humans.

## Introduction

According to the World Health Organization [1], the most prevalent disability-related condition in the world for adults older than 70 years is age-related hearing loss (ARHL). Age-related hearing loss refers to a degenerative process of aging resulting in a progressive bilateral hearing loss [2]. Globally, the prevalence of moderate to severe hearing loss increases exponentially

**Funding:** The author(s) received no specific funding for this work.

**Competing interests:** The authors have declared that no competing interests exist.

with age, rising from 15.4% in people aged in their 60s to 58.2% among the group aged more than 90 years. In the USA the sharpest rise of prevalence of hearing loss occurs in aged more than 80 years. The causes of ARHL are multifactorial and influenced by genetic factors, pre-existing ear conditions, chronic diseases, noise exposure, use of ototoxic medication and life-style. These factors determine the time of onset and development of degenerative changes in the inner ear and higher centers and thus have a great impact on the ability of processing and discriminating acoustic signals.

The inferior colliculus (IC) is a key nucleus in the central auditory pathway that integrates acoustic stimuli from the right and left cochleae and is involved in the localization of sound processing (binaural hearing) [3]. In addition, the IC is regarded responsible for sound recognition in noise [4]. Both sound localization and sound recognition in noise deteriorate with age [5–7]. Fjell and Walhovd [8] found that the brain shrinks in volume in healthy aging, while the ventricular system expands. The shrinking brain is related to shrinkage of neurons, reductions of synaptic spines and lower number of synapses rather than to neuronal loss. The length of myelinated axons is greatly reduced and reductions in specific cognitive abilities are observed in healthy aging. Some brain areas are declining linearly from early in life, whereas others continue to increase in volume well into middle adulthood before eventually beginning to shrink in the later part of life.

Gleich, Netz and Strutz [9] found in histological cross-sections of the IC in gerbils, that the cross-section area of the IC was reduced by 13.1% in old compared to young gerbils. This finding prompted us to investigate, if any age-related variation in size of the human IC exists.

MRI has been described as a method to quantify the volume or thickness of specific brain structures in vivo, yielding a window into the human brain during aging [8]. Haehner et al. [10] were able to reliably evaluate the volume of the olfactory bulb in 20 patients with the help of magnetic resonance imaging. Also, they analysed, if the volume of the olfactory bulb is correlated to olfactory function. As MRI is being routinely used in our hospital for many other reasons, we wanted to ascertain that this non-invasive technique can be used confidently for determining the volume of the IC.

Since Broca's discovery in 1865 [11], that speech is produced in the left hemisphere in most right-handed subjects, studies on structural and functional asymmetries of the brain have become of increasing interest. Wernicke described in 1874 [12] the sensory speech center also being in the left hemisphere in most right-handed people. Handedness as another example for asymmetry of the brain, has the consequence of increased training of one side resulting in shorter reaction times and increased fine motor efficacy. Hemispheric lateralization also enables parallel processing of complementary information within the two hemispheres, reducing cognitive redundancy and increasing overall efficacy [13]. This lateralization is manifest in systematic differences of the right and left volumes of some brain regions [14] especially, it has been shown that language-relevant areas are often larger in the left than in the right hemisphere [15, 16]. In order to find out, if lateralization also applies to the IC, it was intended in this study to compare left and right sides.

In animal models it has been shown that impaired auditory input (deprivation) and/or advanced age can affect the size and number of neurons as well as the volume of auditory brainstem nuclei [17–28]. Thus, a structural change can follow a functional deficit. Therefore, it would also be interesting to know, if a correlation between volume size of IC and hearing performance can be detected.

The questions to be answered by the present study were:

1. Is it possible to determine the volume of the human inferior colliculus by voxel based analysis of routinely performed MRIs?

2. Is there any age-related variation of the volume of the inferior colliculus?

3. Is there any difference in volume between left and right inferior colliculus?

4. Does the volume of the inferior colliculus correlate with existing hearing loss of the patients?

This study was able to answer questions 1–3. A definite answer for question 4 was not achieved.

## Materials and methods

For this study a retrospective search in the data of the Regensburg ENT department was done. The study was approved by the Institute Ethics Committee of the University Regensburg: 18-1214-104.

With reference to the ethics commission decision (hints, point 3) a patient's consent was not necessary, because the study was purely retrospective. The MRI-data for this study were acquired retrospectively from the data bank of the radiology department. Also, the data for the hearing status were acquired retrospectively from the data bank of the audiology department. All data were fully anonymized before access on 12.12.2018. Therefore, the study did not involve any contact with patients or test-subjects. There were no clinical interventions necessary on patients or test-subjects and data collection did not go beyond the evaluation of the medical records and their attachments. Also, the research was carried out without the use of any body materials.

Appropriate cranial MRI data sets (Siemens Magnetom Sola 1.5 Tesla) consisting of 160 sagittal slices with 256x256 voxels were identified. Data sets were only included, if the patient also had an available hearing test in the form of Brainstem evoked response audiometry (BERA) or pure tone audiogram (PTA) and if the time interval between MRI and the performance of the hearing test was less than 6 months. In two patients, who had obtained 2 or 3 MRIs respectively, only the first MRI was included in the present analysis resulting in 123 data-sets meeting the described requirements. The age range of the patients at the time of the MRI was 3 months to 84 years and included 72 males and 51 females.

Screenshots of the MPRAGE MRI image layers were transferred from the Syngo Imaging Software (Siemens) as a stack of images into ImageJ (NIH, ver. 1.43r), and adequately scaled. The "orthogonal view" mode was used to identify the IC in different projections. The corpora quadrigemina consist of the inferior (more caudal) and superior (more rostral) colliculi and are elevations of the tectum forming the dorsal surface of the midbrain. As illustrated in the original orthogonal MRI views in Fig 1 (left side), the surface of the IC is well defined and clearly recognizable. The body of the IC below its surface appears as a brighter area of the tectum. Fig 1 (right side) illustrates how voxels within these borders of the IC were manually labelled as "Region of Interest" to determine the number of voxels belonging to the left and right IC.

The IC-volume was calculated by multiplying the number of the IC-voxels with the voxel-volume which was derived for each image-stack from "slice thickness" (0.7–1.1mm) and "pixel spacing" (0.78–1.02mm) recorded in the corresponding DICOM file.

The PTAs are the standard for diagnosing auditory loss in children who are capable of responding appropriately to the examiner. PTA was used in 67 children and adults and determined air-conduction thresholds using 9 frequencies (125Hz to 8kHz). The BERA was used in 56 young children (aged 3 months to 4 years) in whom a PTA could not be reliably measured, and at least 3 frequencies were included (500-1000Hz, 2kHz, 5kHz). In some deaf children over 1 year of age, a BERA had been performed in the course of cochlear implant diagnostics.

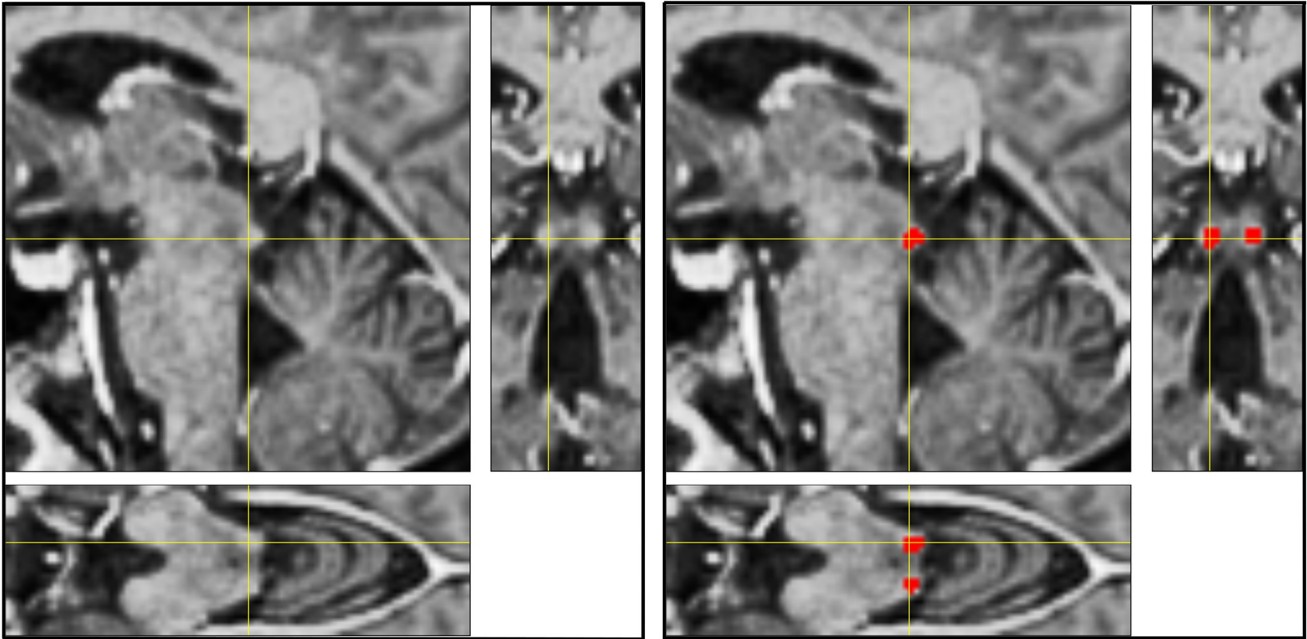

**Fig 1.** Left side: *top left*: sagittal, *bottom*: axial and *top right*: coronal view of ICs. Yellow crosslines marking IC. Right side: Voxels of the ICs are marked in red as regions of interest.

To obtain frequency specific information the response to short tone bursts had been recorded which were used to evaluate the hearing status. For measuring the interpeak-latences I-V click stimuli had been used. If there was more than one threshold determination available, the one which was in closest chronological proximity to the MRI was used.

For each ear, a mean value of the pure tone audiogram thresholds from all measured frequencies was calculated for air conduction and an analogue classification was applied for the BERA data in young children. The mean hearing loss of each patient was classified based on the definition of the guidelines of the World Health Organization [29]. To reduce the number of groups and increase the number of cases within each group for statistical analysis, we defined 3 categories by combining the grades 0 (no impairment, <25dB) and 1 (slight impairment, 26-40dB) as category 1 (≤40dB), grades 2 (moderate impairment, 41-60dB) and 3 (severe impairment, 61-80dB) as category 2 (41-80dB) and grade 4 (profound impairment including deafness, >81dB) as category 3 resulting in 5 possible groups with regard to hearing loss in both ears:

Class I: both sides <40dB, category 1 (Cat 1/1), **normal to mild hearing loss** on both sides,

Class II: <40dB, category 1 on one side and 40-80dB category 2 on the other side (Cat 1/2)

Class III: 40-80dB, category 2 on both sides (Cat 2/2) or <40dB, category 1 on one side and >80dB, category 3 on other side (Cat 1/3)

Class IV: 40-80dB, category 2 on one side and >80dB, category 3 on other side (Cat 2/3)

Class V: >80dB, category 3 on both sides (Cat 3/3), **severe hearing loss to deafness** on both sides

To determine the global hearing loss, the mean threshold was calculated from the thresholds of both ears from tone audiograms.

**Table 1. Distribution of patients in the 4 age groups, mean value of IC-volume from both sides and separately for the left and right ICs.**

| Age group/ years | Mean value age/years, STD | Mean IC volume right/left in mm3 STD | Mean value IC-Volume right/mm3, STD | Mean value IC-Volume left/mm3, STD | Patients Total n = 123 |
|---|---|---|---|---|---|
| 1 (0≤1) | 0.46±0.17 | 17.18±4.46 | 17.15±4.65 | 17.22±4.73 | 40 |
| 2 (>1–10) | 3.64±2.42 | 23.31±7.84 | 23.97±8.79 | 22.65±7.38 | 35 |
| 3 (>10–60) | 38.33±16.81 | 32.00±7.71 | 31.21±8.26 | 32.79±8.41 | 33 |
| 4 (>60) | 70.53±6.84 | 25.95±5.23 | 25.08±6.44 | 26.83±5.33 | 15 |

STD = standard deviation

## Statistics

Appropriate non-parametric tests (Kruskal-Wallis Test, Mann-Whitney-U-Test, Wilcoxon Test, Fisher Exact Test; IBM SPSS Statistics 25) were used for the statistical analysis. A value of $p < 0.05$ was considered to be statistically significant.

## Results

### Volume of ICs in relation to age (Table 1)

The 123 patients were divided into 4 age groups: **group 1**: ≤1 year; **group 2**: 1year to ≤10years; **group 3**: 10 years to ≤60yrs; **group 4**: >60yrs.

Table 1 and Fig 2 show that mean IC volume varied with, age, increasing from ≈17mm$^3$ in the youngest group to ≈23mm$^3$ in children and young adults and to ≈32mm$^3$ in adults up to an age of 60 years. In the oldest group (>60years) mean IC volume was only ≈26mm$^3$ and thus ≈20% smaller than in younger adults.

A Kruskal-Wallis test evaluating the effect of age group on IC volume showed a highly significant effect of the age-group on the mean IC-volume ($p < 0.001$).

Subsequent pair-wise Mann-Whitney-U tests showed that the mean IC volume in babies (≤1 year) was significantly smaller than in the other 3 age groups ($p < 0.001$). Comparing the IC-volume in children (<1 to 10 years) revealed a significantly smaller volume than in adults (>10–60 years; $p < 0.001$), but missed the significance criterion of $p ≤ 0.05$ for the comparison with the oldest group (>60, p = 0.086). The IC–volume in adults (10-≤60 years) was significantly higher than in both younger ($p < 0.001$) but also in the old (>60 years; p = 0.009) groups. These data demonstrate a growth and increasing IC volume after birth reaching a maximum in adulthood and a subsequent shrinkage or loss of volume in old humans over 60 years.

### Right and left IC volume in relation to age

Fig 2 shows the mean IC volume of the right (red triangles) and left (blue diamonds) IC as a function of age (on a logarithmic age scale) for the 123 individuals analyzed.

The mean age and volume are shown for the right IC of the 4 age groups by the larger filled red triangles, connected by a red and for the left IC by the larger blue diamonds connected by a blue line to emphasize the effect of age on IC volume.

The data show no systematic difference in the IC volume between both sides. A Wilcoxon paired sample test comparing left and right IC volume of the 123 subjects found no systematic significant difference or systematic asymmetry between both sides (p = 0.865).

### IC-volumes regarding gender (Fig 3)

The present sample consisted of 51 data-sets from female and 72 from male patients (n = 123). Neither a general comparison of the IC-volume from the 72 males and the 51 females found a

significant difference (MWU, p = 0.312), nor did the comparison in the 4 separate age groups of babies, (≤ 1 year, 21 males, 19 females, MWU, p = 0.728), children (1 to 10 years, 19 males, 16 females, MWU. p = 0.659, adults (>10 to 60 years, 21 males, 12 females, MWU, p = 0.671) and old (60 years, 11 males, 4 females, MWU, p = 0.412) individuals reveal a significant difference between genders.

The group mean data suggest a similar age dependent variation of IC-volume in both genders and Kruskal Wallis tests revealed a significant effect of age in males (N = 72, p<0.001) and females (N = 51, p<0.001). Subsequent pair-wise Mann-Whitney U tests confirmed a significant increase of IC-volume from babies (≤1 year) to children (>1 to 10 years, males p = 0.002, females p = 0.026) and from children (>1 to 10 years) to adults (>10 to 60 years; males p = 0.004, females p = 0.001) in both genders. Despite a similar reduction of IC-volume by ≈6mm$^3$ from adults (>10 to 60 years) to the old group (>60 years) in both genders, this

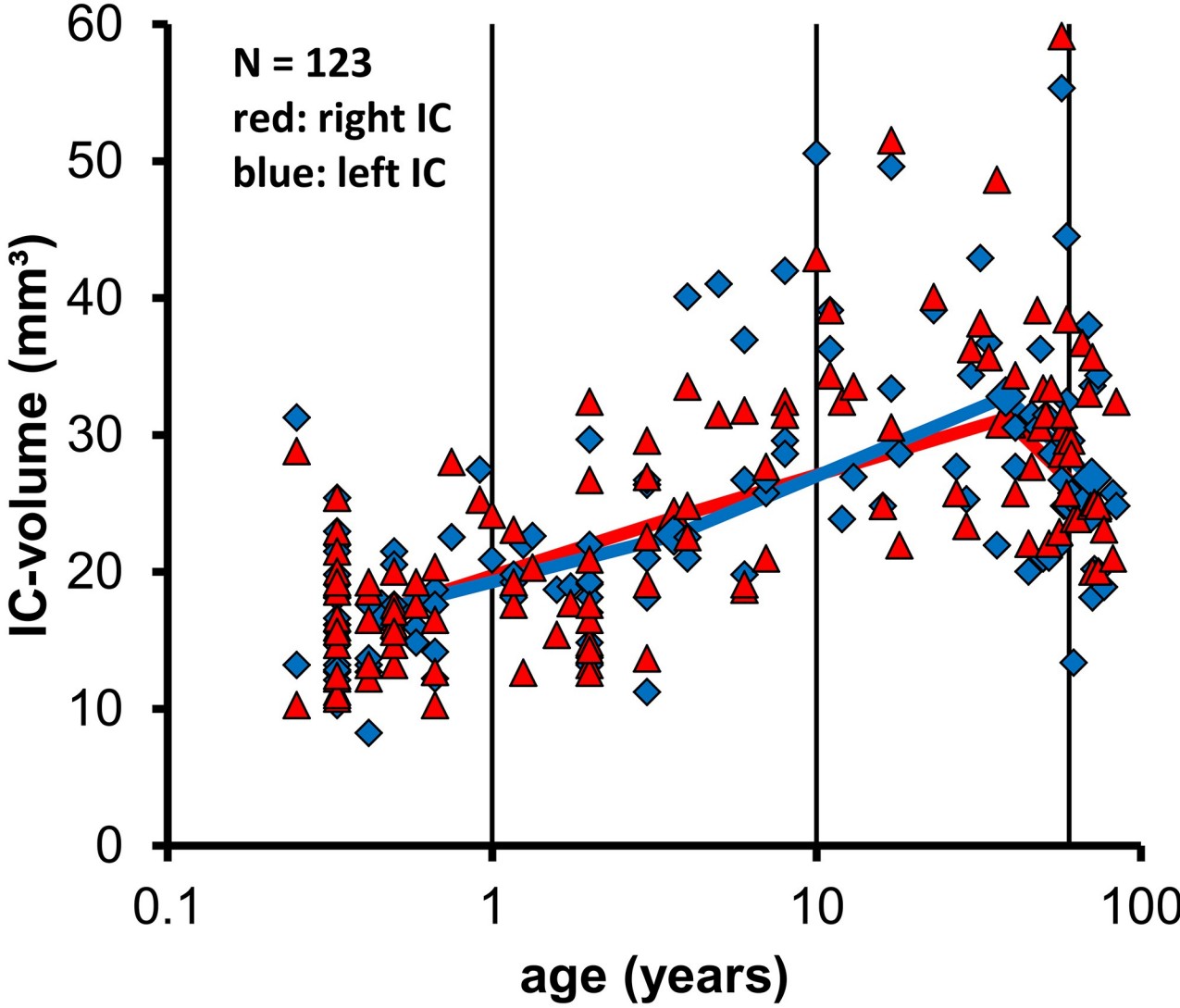

**Fig 2. The volume of the right (red) and left (blue) IC are shown as a function of age on a logarithmic scale.** Means of age groups are indicated by the larger symbols and connected by the red and blue lines. These data show no systematic asymmetry between the volume of the left and right IC and a similar variation of volume as a function of age on both sides. The vertical lines indicate age groups.

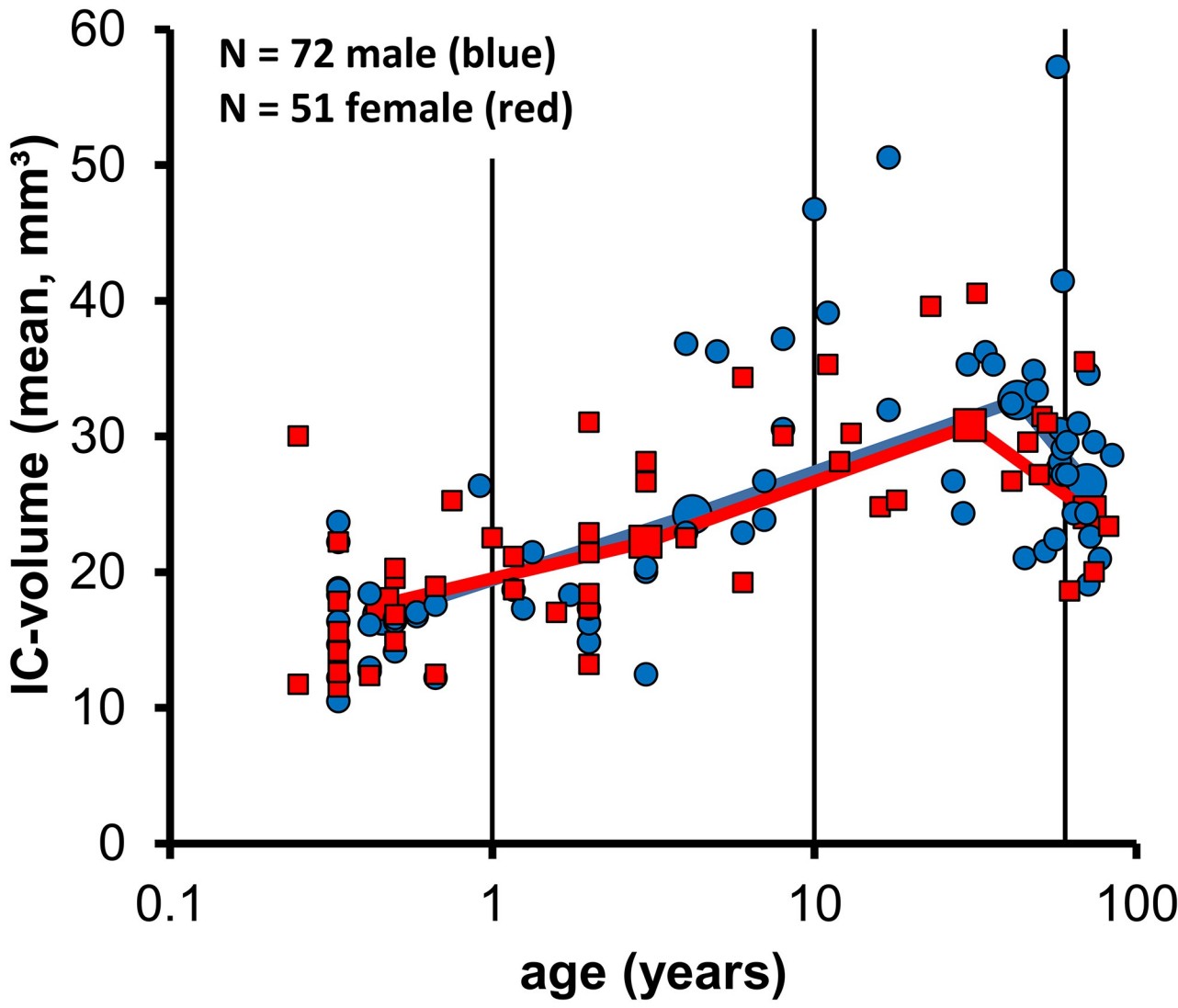

**Fig 3. The small blue circles show the IC-volume (mean of left and right) from 72 males and the small red squares from 51 females as a function of age.** The larger symbols show the means of the 4 age groups from males and females connected by a blue (males) and red (females) line. The data show a considerable inter-individual variability of IC-volume, a wide overlap and no systematic difference of the data distribution in both genders.

difference was significant in males (p = 0.046) but not in females (p = 0.103). However, the small sample size in aged females (N = 4) limits the meaningfulness of conclusions related to IC-volume in old females.

In summary, the present data show no evidence for a difference in IC volume between males and females.

## IC-volumes in relation to hearing loss

Fig 4 illustrates, that in our retrospective sample of MRIs in which also BERA and audiogram data were available, the degree of severe bilateral hearing loss was high in young patients below 10 years (30 of 75 individuals) while it was very low in the adult and aged sample over 10 years of age (2 of 48 individuals). On the other hand, the proportion of individuals with bilateral

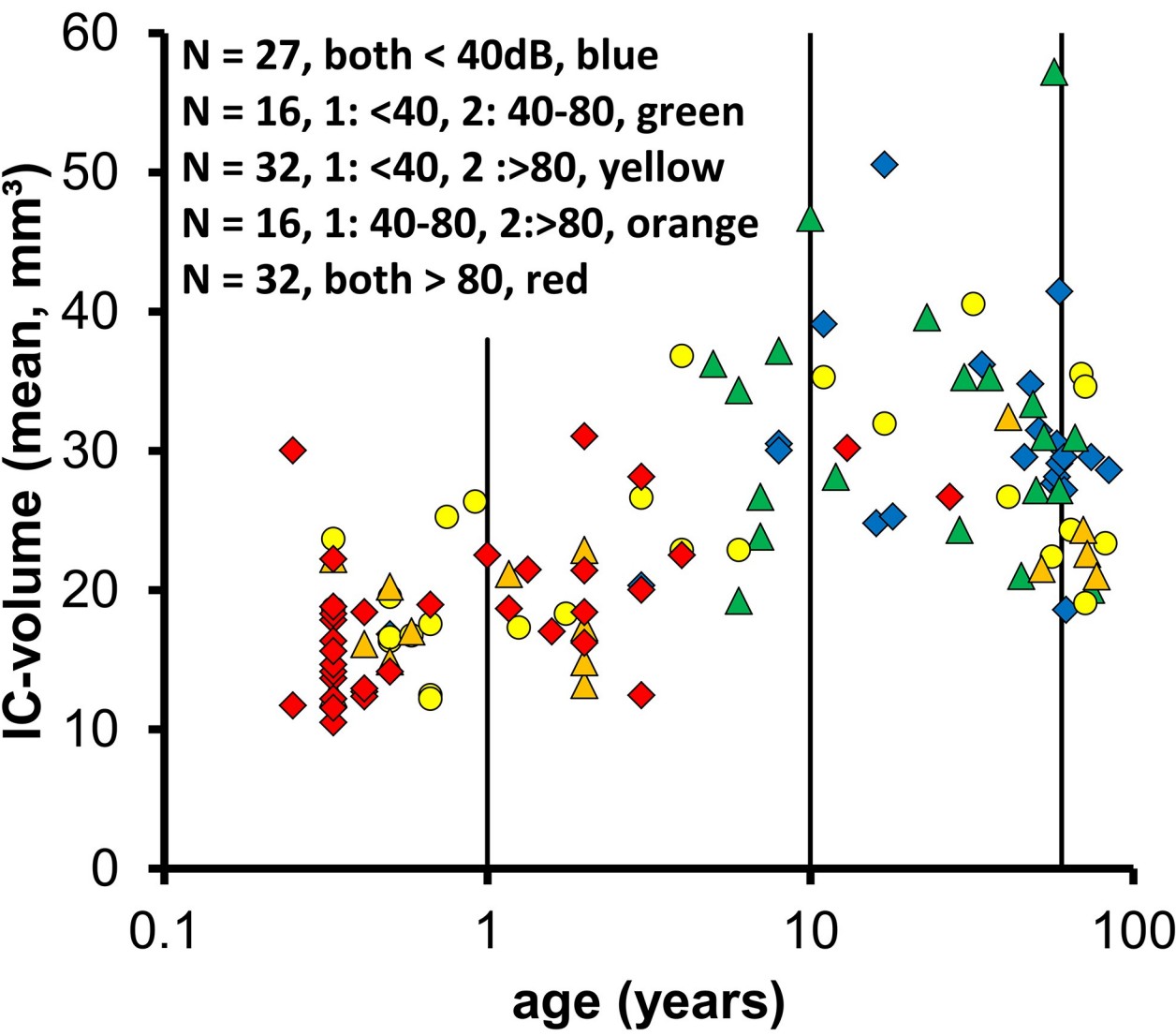

**Fig 4. The diagram shows the mean IC-volume of both sides as a function of age with the degree of hearing loss as defined in the methods from class 1 (normal to moderate in both ears) to class 5 (severe, > 80 dB in both ears) coded by differently colored symbols.**

near normal hearing was low in patients below 10 years (4 of 75) while it was relatively high in the older group (18 of 48). Due to the age dependent variation of IC-volume this sample with a high proportion of hearing loss in young and a low proportion in the older individuals hampers the analysis of the effect of hearing loss on IC-volume.

In the present data set, the pattern of hearing loss in the adult (10–60 years) and old (<60 years) groups appears not obviously different, however, a close look reveals that the proportion of individuals with a high degree of hearing loss on one or both sides (red, yellow and orange symbols) is significantly (Fisher Exact test, p = 0.043) lower in adults (10 to 60 years, 7 of 33) as compared to the old (>60 years, 8 of 15) individuals, suggesting that severe hearing loss (at least on one side) may be associated with a reduced IC-volume in aged humans.

Future analyses require the evaluation of IC-volume in adequate samples of normal hearing and hearing impaired individuals from selected age groups to confirm the hypothesis that hearing loss is associated with a reduced IC volume in aged humans.

### Asymmetrical hearing loss and IC-volume

In the present sample of 123 individuals we identified 13 cases with a near normal hearing (loss <40dB) on one side and a severe loss (>80dB) on the other side (hearing class III). Due to the predominantly crossed input from the ears to the IC [30, 31] we define the IC lying opposite to the ear with a high degree of hearing loss as deprived and the IC lying opposite to the ear with a low degree of hearing loss as normal.

The majority (10 of the 13) of the data points lie above the 1:1 line (Fig 5), indicating that the volume of the normal IC (opposite to the ear with low <40dB hearing loss) is on average higher as compared to the volume of the deprived IC (opposite to the ear with >80dB hearing loss). A paired sample Wilcoxon test comparing the volumes of the normal and deprived ICs

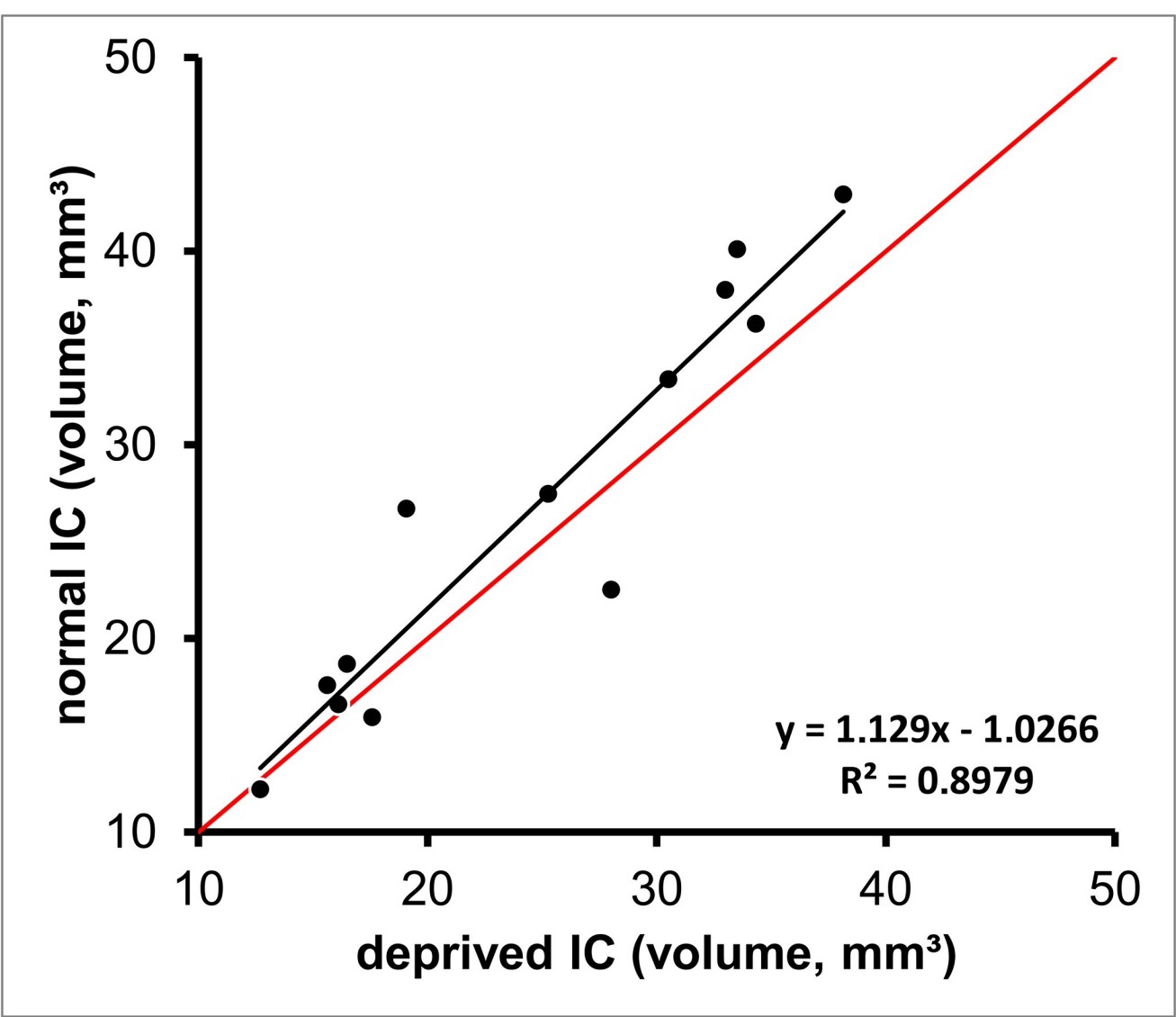

**Fig 5. The diagram plots the volume of the "normal" IC (opposite to the ear with normal hearing <40dB loss; y-axis) as a function of the volume of the "deprived" IC (opposite to the ear with a high degree of hearing loss >80dB; x-axis).** The red line indicates equal IC volume of both ICs. The thin black line shows the linear regression through the data (formula given below the legend).

in the present sample with asymmetrical hearing loss showed that the reduction of the volume on the deprived side in this sample was significant (p = 0.036).

The mean IC volume of this sample of 13 individuals with asymmetrical hearing was 26.79 +/-10.00mm$^3$ for the normal and 24.64+/-8.39mm$^3$ for the deprived IC, indicating on average an 8% reduction of the volume of the deprived IC.

The distribution of data points indicates that the deviation from the 1:1 line increases with increasing volume of the IC: the ratio of the volume on the deprived divided by the volume of the normal IC was 0.97 for the 5 subjects in which the volume of both sides was below 20mm$^3$ and 0.90 for the 8 subjects in which the volume of both sides was above 20mm$^3$.

Due to the age dependent growth of the IC (e.g. Fig 2) the volume difference between the normal and the deprived IC is less pronounced in young (mean age of the 5 subjects with volume of both sides < 20mm$^3$ was 0.6 years) than in older subjects (mean age of the 8 subjects with volume at least on one side > 20mm$^3$ was 17.6 years).

In summary, the present data suggest that in very young subjects (<1year) deprivation is not associated with a shrinkage of IC volume on the deprived side, deprivation rather hampers the normal growth of the IC in children >1year. We also see a significant reduction of IC volume in the age group above 60 years which may be associated with a higher proportion of subjects with a substantial hearing loss (>80dB at least on one side). Unfortunately, we have no information about the onset and time course of the hearing loss in the present sample of 123 subjects analyzed. However, the present analysis demonstrates that the MRI based evaluation of IC volume is suitable to detect relevant effects of age and hearing status. Adequately defined samples of patients with better defined histories of hearing status are required for a more detailed analysis of the relation between the IC-volume and hearing status (e.g. can a hearing aid or CI support a "normal" development).

## Discussion

Age-related hearing loss is a common and global issue, causing a high rate of disability among adults older than 70years [1]. The prevalence of moderate to severe hearing loss rises with age from 15.4% in adults in their 60s to 58% in aged more than 90 years. Especially, speech discrimination in background noise and binaural hearing for locating sound become compromised as a result of ARHL [4]. The inferior colliculus plays a key-role in these functions and each IC receives stimuli from both cochleae supporting this role.

### Method of measuring the IC volume

Macech [32] performed measurements (length, width and height) of the IC on the anterior surface of the brain stem in unfixed cadavers using a slide caliper.

Volume measurements of the IC have been undertaken by Glendenning and Masterton [33] in various mammals including the human, using fixed brain sections. For these fixed sections, shrinkage of the tissues during preparation of the histological material had to be taken into account.

MRI-based measurements for determining volumes of brain-structures have been performed in many previous studies [8, 10, 34–36]. The in-vivo MRI-based measurements have the advantage of not to be compromised by any shrinkage artifacts and should yield more realistic values than measurements based on histological material.

MRI-based measurements of the IC should also be more exact than measurements taken from the surface of the brainstem with a slide caliper, as the extension of the IC into the depth cannot be seen from the surface of the brainstem.

Haehner et al. [10] were able to evaluate the volume of the olfactory bulb in 20 patients reliably with the help of magnetic resonance imaging. The mean volume of the olfactory bulb is stated with 44-57mm$^3$ in their study, which is within the size-range of our measurements of the IC volume in adults (range 21.0 to 57.2mm$^3$, mean: 32.0+/-7.7mm$^3$)

In our study, the IC-volume revealed a remarkable inter-individual variation even within the same age-group. The right and left IC of the individual patients and also the mean values of right and left sides, however, were very similar in volumes. Also, there was no significant difference in mean values of IC-volumes concerning the gender (Fig 3).

In summary, our analysis shows that the IC-measurements derived from the MRI-data are accurate and reliable enough for the study of effects of age and hearing status.

## IC volume related to age

Some basic hearing abilities are well developed at birth, but there is a clear prolonged maturation of auditory development well into the teenage years [37]. Also non-auditory factors like attention, memory and cognition contribute to auditory development. The ability of a system to adapt in response to novel stimuli is a key feature of development throughout the nervous system and is called neural plasticity [38].

One of the main outcomes in this study is the increase in human IC volume from birth into adulthood and the decrease over the age of 60 years.

This suggests that the IC as part of the auditory system with its plasticity also undergoes a maturing development in childhood and a degenerative structural as well as decreasing functional change at old age.

Similarly to our results, Konigsmark and Murphy [39] described in a histological study with formalin fixation and celloidin embedding that the volume of the ventral cochlear nucleus in humans increases during the first decades of life and decreases beyond the fifth decade of life. They reported that this was not due to a loss of neurons, but that there was a marked decrease in packing density with an increased amount of neuropil in middle age. The structures apparently increasing from infancy to middle age, and decreasing in old age, were the myelinated axons in the ventral cochlear nucleus.

The cells of the IC projecting to the auditory thalamus are increasingly surrounded by perineuronal nets with age in the rat model [40]. Perineuronal nets can stabilize synapses and are also important for promoting functional synaptic plasticity [41, 42]. The IC is likely compensating for the age-related changes to GABA A-receptors and the decline of synapses [43] which starts in middle age and significantly rises to a loss of 28% in old age. This compensation appears to work rather well for most of life [44], as deficits to temporal and speech processing typically present only in aged people [45].

It could be possible, that increasing myelination, neuropil and increasing perineuronal nets over lifespan are responsible factors for the increasing IC volume from birth to adulthood and for the compensation of age-related decline in function.

Our findings of a reduced IC volume beyond the age of 60 years in humans can be considered in accordance with the observation of a significant age-related decline in the total number of Gaba-ergic neurons in the human IC [3], a decreased number of GABA immunoreactive neurons and decreased concentration of GABA in the aged rat [4] and a shrinkage of the IC without a loss of GABAergic neurons in gerbils [9]. Despite some species-specific variability, these structural age dependent changes of the IC might affect/impair auditory function especially the quality of temporal and spatial processing of sound.

Deprivation, especially during critical periods affect the development of nuclei in the ascending auditory pathway [46], demonstrating that a functional deficit can lead to a

structural change. Independent of the critical periods during development, advanced age is also associated with shrinkage of auditory nuclei [9, 39].

In summary, the results of this study match the hypothesis from the introduction, that an age-related decrease in size of the IC in humans does exist, coinciding with the thought, that structural changes associated with advanced age are associated with a decline in function of the IC. These structural and functional changes found in the IC are possible explanations for age-related hearing loss, especially the decline of spatial and temporal processing of sound as well as hearing deficits in noise in the aged population.

For future research it would be interesting to analyze the ICs of adult patients below the age of 60 years who have had a long lasting, temporally defined hearing loss beginning either during or after the critical period on one or both sides and correlate IC volumes with deficits associated with the accuracy of sound localization.

### Size of the IC in general

Glendenning and Masterton [33] compared the volumes of structures in the auditory system of 53 mammals and found that among the examined mammals, the humans showed the largest absolute size but the smallest auditory system, when taken into relation to their total brain size. In their study, the in vitro volumes of the auditory nuclei were estimated from fixed brain sections and only one animal/human individual was used to represent a species. Because of possible error measurements due to shrinkage of the tissue during the histological processing, it is stressed that the error is minimized by restricting the comparisons to relative instead of absolute volumes. With these considerations in mind, the IC volume determined in one human individual was 65.2 mm$^3$. Our study showed a systematic variation of IC-volume with age (e.g. Fig 3) and in adults (age 10–60 years, N = 33) we found a lower mean IC volume of 32 +/- 7.7mm$^3$. The highest IC volumes found in two individuals in the present study were between 50 and 60mm$^3$. Considering the different methodology of the two studies, the reported volumes appear compatible.

### IC volume in relation to left/right

The analysis of right and left IC volumes could not detect any systematic asymmetry between the two sides (Fig 2). This finding could underline the fact, that the function of the IC is spatial differentiation and localization of sound, both requiring a symmetric comparison of the input provided to the two ears. In our study, no lateralization of the brain in regard to the IC could be shown, unlike in left/right-handedness or for the speech-center in humans for which lateralization is well-known.

It is a constraint of this study, that handedness cannot not be considered in the discussion of the IC volume in relation to right/left sides (lateralization of hearing), as there were no records available.

### Difference in IC volume depending on hearing status

There have been reports on contralateral hearing loss after lesions of the IC [47, 48] as well as tinnitus and impaired sound localization in the contralateral hemisphere without hearing loss [49] after an intracerebral bleeding in the central nucleus of the IC.

In gerbils, unilateral conductive hearing loss induced a decrease of deoxyglucose uptake in the ipsilateral cochlear nucleus and the contralateral IC [50]. Similarly, Speck et al. [51] described a significant decrease in glucose uptake in the contralateral IC in asymmetrical hearing impaired individuals undergoing 18F-FDG PET imaging as a measure of neuronal activity.

Gleich and Strutz [25] studied the effect of unilateral conductive hearing-deprivation during the critical period in gerbils. They found that the volume of the anteroventral cochlear nucleus in 6 months old gerbils was reduced by 20% when deprivation began at 2 weeks of age but was not affected when deprivation started in 3-months old animals. This clearly showed a structural change following a functional deficit that occurred during the critical period.

We found 13 patients with an asymmetric hearing loss. Due to the predominantly crossed input from the ears to the IC [30, 31] we defined the IC lying opposite to the ear with a high degree of hearing loss as deprived and the IC lying opposite to the ear with a low degree of hearing loss as normal. In these 13 patients a significant reduction of the volume on the deprived side (Wilcoxon test, $p = 0.036$) could be detected. Our results indicated on average an 8% reduction of the volume of the deprived IC suggesting that hearing loss is associated with a reduced IC-volume.

One of the shortcomings of this study was the small sample size in >60year-olds with a severe hearing impairment and also in infants with normal hearing. This could have been one of the reasons why a general correlation between hearing loss and volume of ipsi- and contra-lateral IC could not be detected in the present sample.

The findings also support the assumption, that hearing loss may have to be long-lasting before it leads to a change in IC volume and that only substructural and functional changes might be present right after the onset of hearing loss.

Since this exploratory retrospective study was, to our knowledge, the first systematic trying to correlate the relation of IC-volume with other factors like age or hearing status, these data can provide the basis to estimate the required sample size for more detailed future studies.

## Conclusion

It can be concluded from the results of this study that it is possible to use a voxel-based method on routine clinical MRI stacks to determine the volume of the human inferior colliculus.

The volume of the inferior colliculus shows an age-dependency. There is a growth in IC volume from infancy until adulthood and a significant decrease in patients over the age of 60 years.

Left and right inferior colliculi do not show any systematic asymmetry in volume and there is no difference in volume between females and males.

In the group with asymmetric hearing (n = 13) a significant reduction of the volume on the deprived side (p = 0.036) was found. The proportion of subjects with severe hearing loss at least on one side was significantly higher in the old (>60 years) as compared to younger adults (10 to 60 years), suggesting that severe hearing loss may be associated with a reduced IC-volume in aged humans.

## Supporting information

**S1 Table. Raw data.** The excel sheet shows gender and age/years rounded to whole numbers of the patients (columns B&C), the DICOM attributes (columns D-H) used for determining the voxel volume (column J), the number of IC voxels (column K,L), IC volume (columns M, N) and the results of the hearing tests (columns Q-AR, frequencies where the highest output of the equipment did not elicit a response and no threshold could be determined are indicated by "-", missing data points are indicated by "o").
(XLSX)

## Acknowledgments

We thank Steven Marcrum for help with improving the English language.

## Author Contributions

**Conceptualization:** Otto Gleich.

**Data curation:** Pingling Kwok, Otto Gleich, Peter Koch, Gudrun Schenkl, Nina Koch.

**Formal analysis:** Otto Gleich, Gudrun Schenkl.

**Investigation:** Pingling Kwok, Otto Gleich, Peter Koch, Gudrun Schenkl.

**Methodology:** Pingling Kwok, Otto Gleich.

**Project administration:** Christopher Bohr.

**Resources:** Christopher Bohr.

**Supervision:** Pingling Kwok, Otto Gleich, Christopher Bohr.

**Visualization:** Christopher Bohr.

**Writing – original draft:** Pingling Kwok, Otto Gleich.

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
