## [Decision Letter · Decision Letter 0]

5 Sep 2024

PONE-D-24-15625Measurement of inferior colliculus volume  based on MRI image stacks and its relationship with age and hearing statusPLOS ONE

Dear Dr. Kwok,

Thank you for submitting your manuscript to PLOS ONE. After careful consideration, we feel that it has merit but does not fully meet PLOS ONE’s publication criteria as it currently stands. Therefore, we invite you to submit a revised version of the manuscript that addresses the points raised during the review process.

We look forward to receiving your revised manuscript.

Kind regards,

Ryota Sakurai, Ph.D.

Academic Editor

PLOS ONE

2. In the online submission form you indicate that your data is not available for proprietary reasons and have provided a contact point for accessing this data. Please note that your current contact point is a co-author on this manuscript. According to our Data Policy, the contact point must not be an author on the manuscript and must be an institutional contact, ideally not an individual. Please revise your data statement to a non-author institutional point of contact, such as a data access or ethics committee, and send this to us via return email. Please also include contact information for the third party organization, and please include the full citation of where the data can be found.

4. Please include a caption for figure 5.

Additional Editor Comments:

Two reviewers made some comments on this paper. Please respond to all these comments.

Reviewers' comments:

Reviewer's Responses to Questions

**Comments to the Author**

1. Is the manuscript technically sound, and do the data support the conclusions?

Reviewer #1: Partly

Reviewer #2: Partly

2. Has the statistical analysis been performed appropriately and rigorously? 

Reviewer #1: Yes

Reviewer #2: No

3. Have the authors made all data underlying the findings in their manuscript fully available?

Reviewer #1: Yes

Reviewer #2: No

4. Is the manuscript presented in an intelligible fashion and written in standard English?

Reviewer #1: Yes

Reviewer #2: Yes

5. Review Comments to the Author

Reviewer #1: Initially, I would like to emphasize the relevance and significance of this subject matter to the field of hearing science.

I will provide a few suggestions and recommendations below to enhance the manuscript.

You mentioned that all patients have PTA or BERA. However, it is crucial to specify the type of BERA that was employed, as the BERA-click does not provide data that is comparable to PTA. Therefore, examine this item and substitute the procedure name for BERA-specific frequency.

PTA – indicating the values and equipment that are used.

The BERA frequency specifies the values and equipment that are employed. Before the specified frequency was reached, BERA was executed–click?

Why was 5kHz employed? This frequency is not of great significance within an audio battery.

It is strongly recommended that auditory loss be classified in accordance with the guidelines of the World Health Organization (2019). And that the categories be divided by this classification.

I would have appreciated the opportunity to review a prior study that examined the number of participants who would be considered reliable for this study.

Note that the size of the letters is greater than the remainder of the manuscript on page 10, lines 182 to 184.

It is essential to mention in a specific section of the manuscript that the PTA is the standard for diagnosing auditory loss in children who are capable of responding appropriately to the elderly.

And that the BERA is an extremely useful assessment instrument for quantifying auditory thresholds in individuals who are unable to provide good responses on the PTA. It is particularly the primary diagnostic method for lactating women, infants, and children.

In the discussion, the first sentence indicates that hearing loss is more prevalent after the age of 70. I suggest reading additional articles, as there is already evidence that it occurs more frequently today. Additionally, I suggest that the discussion regarding prevalence and incidence be more in-depth, including, for instance, the fact that adolescents and children tend to lose their hearing even more rapidly as a result of the use of hearing aids. The initial paragraph is extremely rudimentary and inadequate.

In terms of the IC volume, it was determined that two individuals exhibited values that were higher. I suggest that they draw conclusions about this. What was the profession of these individuals? Were they musicians? In conclusion, they must better package this item.

Furthermore, when discussing the IC volume in relation to left/right, they also mentioned left/right-handedness very appropriately. However, did you analyze this variable? It would be an extremely significant point in this study. Given the impossibility of this analysis, as it is a retrospective study, I suggest that it be included as a constraint on the study.

I suggest that the study's limiting factors, which encompass a multitude of other factors, be updated, and that new studies be proposed.

Furthermore, it is not possible to generalize and assert that there is no symmetry, which would imply that this was observed in the current study with a reduced number of participants. The same applies to the other apprehensions. This study must be replicated with an appropriate number of participants, as well as more clearly defined evaluation procedures.

Reviewer #2: I thank the editors of PLOS One for the opportunity to review this article titled “Measurement of inferior colliculus volume based on MRI image stacks and its relationship with age and hearing status”. In this article, the authors report a study into the association between inferior colliculus volume (measured retrospectively using MRI) and patient age, sex, and hearing status. The article reviews the prevalence and impact of age-related hearing loss, the role of the inferior colliculus, and the phenomenon of lateralisation in the brain. The authors address the knowledge gap of whether there is any lateralisation of the IC in humans, and how IC volume correlates with age and hearing loss. The article concludes that IC volume is age dependent and symmetrical, and speculates that IC volume may be associated with age.

My major concerns in relation to the article are regarding the choice of participant groups and the statistical comparisons made. The group comparisons performed do not seem to be the most intuitive and as such the reader is left wondering whether further tests were performed but not included in the article as they were non-significant. More justification needs to be given and more direct tests need to be performed and reported.

I worry a little about the slight over-statement in the abstract “suggesting that severe hearing loss may be associated with a reduced volume of the inferior colliculus in aged humans”. This seems much diluted in the article, whereby the and of the introduction states that “a definite answer for question 4 was not achieved”. I feel that the devinitiveness of the study findings should be presented consistently throughout the article, and I would recommend toning-down the speculative claims in the abstract as they are not representative of the article as a whole.

It would be useful in the introduction to briefly define age-related hearing loss. What sort of thresholds are considered significant? Who’s definition is this? You can then refer to this in the methods section when defining loss in the context of the study.

Why were undeveloped children (3 months up) included in the study? Would you not expect that the IC would still be increasing in size at that point? Do you have any evidence for a maximum IC size or an age at which the brain has reached its adult size in the majority of people? I later see you used 10 years as the cut-off between childhood and adult – is there any evidence for the IC not increasing in size beyond 10 years old?

How was the edge of the IC determined? Please describe all methods used – was there a tissue contrast visible? Were some assumptions made about the internal anatomy of the brainstem?

The first mention of BERA (line 111) is not defined – but it is later.

Why was 6 months between audiometry and MRI used as the cutoff? Do you have any evidence supporting this or supporting not allowing a longer period of time between assessments? Broadening the period may introduce more noise into the associations but would it give you many more datapoints? Could you do a power calculation to see how many you would expect to need? If this could be done on a subset of the data, you may even be able to complete your new analysis with the remainder of the dataset.

Please pull the MRI scanning parameters together more helpfully. You state that all images had the same matrix size (256*256*160) but later imply that the voxel sizes varied in all three dimensions. Is this the case? Were all scans definitely MPRAGEs or did the contrast and/or readout vary? Did all scans have square in-plane voxels? Why did the voxel size vary (I understand that this data is all from a single site – correct me if I am wrong)? Was the protocol updated at a particular timepoint? Or do different protocols use different parameters – is it possible that the chosen protocol may be associated with the investigation of any other pathology (e.g., stroke, trauma, tumour)?

Were you able to see the reason that the MRI was requested? Was that associated in any way?

I had to re-read lines 130-132 a few times to understand what you meant. Please reword to clearly state that PTA was used in 67 cases and BERA in the remaining 56.

Why pool all 10-60 year olds into one large group? Surely it would be more useful to divide this group down? Please justify your methodology. Especially in light of your statement (lines 312-13) that maturation proceeds into the teenage years.

You mostly refer to gender throughout but male and female are sexes (man and woman are genders). It’s fine to say sex as this is medical records you’re dealing with. It’s more important to be consistent.

I disagree with the statement (lines 233-237) that the the proportion of individuals with a high degree of hearing loss on one or both sides being significantly lower in adults (10 to 60 years, 7 of 33) as compared to the old (>60 years, 8 of 15) individuals suggests that severe hearing loss (at least on one side) may be associated with a reduced IC-volume in aged humans. If that were the case you would be able to test directly for the association between PTA/BERA and IC volume.

Your within-subject asymmetric hearing vs asymmetric IC volume finding is much more interesting and compelling.

I don’t fully understand why you have performed the comparisons you have, used the group definitions you have, and omitted what seems to be the most obvious tests. Did you perform more tests with non-significant outcomes? All tests need to reported and if necessary multiple comparisons need to be corrected for.

6. PLOS authors have the option to publish the peer review history of their article (what does this mean?). If published, this will include your full peer review and any attached files.

Reviewer #1: No

Reviewer #2: No

---

## [Author Response · Author response to Decision Letter 0]

9 Oct 2024

PONE-D-24-15625 Response to Reviewers

Measurement of inferior colliculus volume based on MRI image stacks and its relationship with age and hearing status

Dear Prof. Sakurai and dear reviewers,

thank you for your constructive recommendations which are most helpful in improving this manuscript.

Academic Editor:

1) We have adapted the format of our manuscript to the PLOS ONE´s style requirements. 

We have stated the references in Vancouver style. Also, we have added supporting information (all raw data under "S1 Table"); the caption for this has been added at the end of the manuscript.

2) In the online submission form you indicate that your data is not available for proprietary reasons and have provided a contact point for accessing this data.

The data has now been made available in the supplementary information "S1 Table".

3) Please amend either the title on the online submission form (via Edi Submission) or the title in the manuscript so that they are identical.

This has been done.

 4. Please include a caption for figure 5.

The caption for figure 5 is in the lines 252-255 of the manuscript.

Reviewer # 1

You mentioned that all patients have PTA or BERA. However, it is crucial to specify the type of BERA that was employed, as the BERA-click does not provide data that is comparable to PTA. Therefore, examine this item and substitute the procedure name for BERA-specific frequency.

For the present exploratory retrospective analysis of the relation of hearing status and IC volume we noticed early during the study the heterogeneity of the subjects with regard to age and hearing status (e.g. Fig. 4). Thus, we focused on a coarse classification of hearing loss that can be achieved by evaluating standard clinical PTAs and BERAS performed in the audiology section of the Regensburg ENT clinic. 

To obtain frequency specific information we recorded in Regensburg the response to short tone bursts which were used to evaluate the hearing status. For measuring the interpeak-latences I-V click stimuli were used. 

We have added a sentence concerning this issue in lines 137-139 of the manuscript.

PTA – indicating the values and equipment that are used.

See Raw data (S1 Table) that are now provided in “supplementary data” and give the thresholds obtained at the PTA test-frequencies and for the BERA stimuli.

The equipment used for PTA: Madsen Astera, Co. Otometrics/Münster-Germany

The equipment used for BERA: Navigator Pro, Bio-Logic/Natus, Model 580-NVBOX1156A, Worthing, UK.

The BERA frequency specifies the values and equipment that are employed. Before the specified frequency was reached, BERA was executed–click?

To obtain frequency specific information we recorded in Regensburg the response to short tone bursts which were used to evaluate the hearing status. For measuring the interpeak-latences I-V click stimuli were used. 

Why was 5kHz employed? This frequency is not of great significance within an audio battery.

For the BERA values in this study, the Navigator Pro-machine was used which had been preset for the examination with click-stimuli using the upper frequency limit at 5kHz. 

In our newer BERA equipment (Eclipse by Interacoustics, Middelfart, Denmark) the upper frequency limit for clicks is now 4kHz.

However, as this is a retrospective study, the examination frequency cannot be changed to 4kHz. Consequently, high frequency loss in the BERA measurements was determined in the majority of earlier data by 5kHz and in more recent measurements with a 4kHz setting. 

It is strongly recommended that auditory loss be classified in accordance with the guidelines of the World Health Organization (2019). And that the categories be divided by this classification.

Our classification is in close accordance with the classification mentioned above. It combines the grades 0 (no impairment, <25dB) and 1 (slight impairment, 26-40dB) as category 1 (normal to mild hearing loss) and grade 4 (profound impairment including deafness, >81dB) as category 3 (severe hearing loss or deafness) as listed in lines 148-154. 

I would have appreciated the opportunity to review a prior study that examined the number of participants who would be considered reliable for this study. 

Since this exploratory retrospective study was, to our knowledge, the first systematic trying to correlate the relation of IC-volume with other factors like age or hearing status, these data can provide the basis to estimate the required sample size for more detailed future studies. 

Note that the size of the letters is greater than the remainder of the manuscript on page 10, lines 182 to 184.

This formatting mistake has been corrected. Due to corrections in the manuscript the lines have shifted to 189-191.

It is essential to mention in a specific section of the manuscript that the PTA is the standard for diagnosing auditory loss in children who are capable of responding appropriately to the elderly.

PTA was used in older children and adults when possible. BERA was used in babies when a PTA could not be reliably measured. In some deaf children over 1 year a BERA was performed in the course of Cochlear Implant diagnostics. 

This we now mentioned in the methods part lines 132-137:

The PTAs are the standard for diagnosing auditory loss in children who are capable of responding appropriately to the examiner. PTA was used in 67 children and adults and determined air-conduction thresholds using 9 frequencies (125Hz to 8kHz). The BERA was used in 56 young children (aged 3 months to 4 years) in whom a PTA could not be reliably measured, and at least 3 frequencies were included (500-1000Hz, 2kHz, 5kHz). In some deaf children over 1 year of age, a BERA had been performed in the course of cochlear implant diagnostics.

In the discussion, the first sentence indicates that hearing loss is more prevalent after the age of 70. I suggest reading additional articles, as there is already evidence that it occurs more frequently today. 

The National Institute on Deafness and Other Communication Disorders states in their Quick Statistics about Hearing, Balance & Dizziness Report which was updated Sept. 20, 2024, that among adults ages 20-69 who report 5 or more years of exposure to very loud noise at work, about 18% have speech-frequency hearing loss in both ears. About 5% of adults ages 45-54 have disabling hearing loss. The rate increases to 10% for adults ages 55-64. 22% of those ages 65-74 and 55% of those who are 75 and older have disabling hearing loss.

These numbers do not differ so much from the ones stated in our discussion. 

Another study from 2010 looked at hearing thresholds from 1999-2004 and compared them with thresholds from 1959-1962 ( Hoffman HJ et al. Americans hear as well or better today compared with 40 years ago: hearing threshold levels in the unscreened adult population of the United States, 1959-1962 and 1999-2004. Ear Hear 2010 Dec;31(6):725-34.) They found that prevalence of overall hearing impairment was significantly lower in 1999-2004 than in 1959-1962.

However, the population worldwide is becoming older and, therefore, age-related hearing loss can be found more frequently than in the past. 

Additionally, I suggest that the discussion regarding prevalence and incidence be more in-depth, including, for instance, the fact that adolescents and children tend to lose their hearing even more rapidly as a result of the use of hearing aids. The initial paragraph is extremely rudimentary and inadequate. 

The aim of this study was to examine the volumes of the inferior colliculus and relate them with age and hearing loss. A discussion regarding prevalence of hearing loss and the impact of the use of hearing aids on hearing in adolescents would be suitable for a review study but not for this retrospective analysis, which cannot find answers for the changes in prevalence of hearing loss.

In terms of the IC volume, it was determined that two individuals exhibited values that were higher. I suggest that they draw conclusions about this. What was the profession of these individuals? Were they musicians? In conclusion, they must better package this item. 

This is an interesting point and we looked up these two cases.

Both patients with the high IC volumes happened to be deaf children (3 months and 4 years old), so they were certainly not musicians. We could not draw any sound conclusions from the results.

Furthermore, when discussing the IC volume in relation to left/right, they also mentioned left/right-handedness very appropriately. However, did you analyze this variable? 

Unfortunately, we have no records of the handedness. 

It would be an extremely significant point in this study. Given the impossibility of this analysis, as it is a retrospective study, I suggest that it be included as a constraint on the study. 

The following sentence was added to the discussion lines 387-388

It is a constraint of this study, that handedness cannot not be considered in the discussion of the IC volume in relation to right/left sides (lateralization of hearing), as there were no records available.

I suggest that the study's limiting factors, which encompass a multitude of other factors, be updated, and that new studies be proposed. 

This sentence was added at the end of the discussion line 416-418:

Since this exploratory retrospective study was, to our knowledge, the first systematic trying to correlate the relation of IC-volume with other factors like age or hearing status, these data can provide the basis for focused study designs comparing age matched groups with well-defined hearing status and for the estimation of the required sample size for more detailed future studies. 

Furthermore, it is not possible to generalize and assert that there is no symmetry, which would imply that this was observed in the current study with a reduced number of participants. 

A Wilcoxon paired sample test comparing left and right IC volume of the 123 subjects found no systematic significant difference or systematic asymmetry between both sides (p = 0.865). 

We do not state in the manuscript that “there is no symmetry”, rather we describe that the data “show no systematic difference … between both sides.” and this is also reflected in the discussion lines 381 – 386. 

The same applies to the other apprehensions. This study must be replicated with an appropriate number of participants, as well as more clearly defined evaluation procedures.

We agree that our retrospective explorative analysis cannot unequivocally answer many detailed questions. But it provides the basis for the focused design of future studies with adequate estimates of required sample size. 

Reviewer #2

I thank the editors of PLOS One for the opportunity to review this article titled “Measurement of inferior colliculus volume based on MRI image stacks and its relationship with age and hearing status”. In this article, the authors report a study into the association between inferior colliculus volume (measured retrospectively using MRI) and patient age, sex, and hearing status. The article reviews the prevalence and impact of age-related hearing loss, the role of the inferior colliculus, and the phenomenon of lateralisation in the brain. The authors address the knowledge gap of whether there is any lateralisation of the IC in humans, and how IC volume correlates with age and hearing loss. The article concludes that IC volume is age dependent and symmetrical, and speculates that IC volume may be associated with age.

My major concerns in relation to the article are regarding the choice of participant groups and the statistical comparisons made. The group comparisons performed do not seem to be the most intuitive and as such the reader is left wondering whether further tests were performed but not included in the article as they were non-significant. More justification needs to be given and more direct tests need to be performed and reported.

This is an explorative analysis of retrospective data to evaluate the feasibility if IC-volume measurements in routine clinical MRIs. In addition, we attempted to use the resulting volume data to test a number of hypotheses that appeared fairly relevant to us in the context of auditory research: 

- Effect of age 

- Left – right symmetry

- Effect of hearing loss

We have not tested additional hypotheses (e.g. effect of left-right handedness on IC symmetry, or effect of “musicality” on IC volume, or the effect of smoking) because we did not have these data. We did not selectively report “significant” results and “drop insignificant” results, we explored the hypotheses listed above and presented the results based on the available data base. 

I worry a little about the slight over-statement in the abstract “suggesting that severe hearing loss may be associated with a reduced volume of the inferior colliculus in aged humans”. This seems much diluted in the article, whereby the and of the introduction states that “a definite answer for question 4 was not achieved”

We feel that our data “suggest that a severe hearing loss may be associated with a reduced volume in the inferior colliculus in aged humans” (e.g. see Fig. 4 that shows that in the group with an age over 60 years the proportion of individuals with a high degree of hearing loss on one or both sides is significantly higher and IC volume on average reduced as compared to the group of younger adults). 

As suggested by data presented in Fig. 5 that analyzed IC-volume from individuals with a pronounced asymmetrical hearing loss, the Wilcoxon test for the 13 available data points revealed a significantly smaller IC volume for the deprived as compared to the “normal” IC. However, this asymmetry was not apparent in the youngest individuals suggesting that deprivation does not lead to a shrinkage of IC volume in babies but rather may affect the normal age dependent growth. Consequently, we are carefully stating at the end of the introduction, that we cannot generally confirm that the IC volume correlates with hearing loss. Additional data from carefully selected sets of patients with defined age groups and hearing status would be necessary to evaluate the effect of hearing status on IC volume in more detail.

Motivated by your concerns, we have looked again at the 48 adult cases (>10yrs) and examined the correlation between IC volume and average PTA threshold of the contra-lateral ear. Because we obtained 2 related measurements (left and right IC) from 48 independent subjects in the present sample, the analysis was performed separately for the left and the right IC. The corresponding plots are shown in the diagrams below. 

The linear correlation analysis of hearing loss in the contralateral ear as a function of IC-volume for the 48 adults over 10 years of age resulted for both ears in a regression line (left IC: y = -0.711x + 62.013; R² = 0.0246; right IC: y = -1.0015x + 69.305; R² = 0.0841) with a negative slope, suggesting that a small IC-volume tends to be associated with a high hearing loss and vice versa. In the present sample of adults over 10 year of age the effect of hearing loss was significant (p<0.05) for the right, but not for the left IC. The coefficients of determination show that the degree of hearing loss explains roughly 2.5% and 8.4% of IC-volume variation of the left and right IC respectively in this sample. 

I feel that the devinitiveness of the study findings should be presented consistently throughout the article, and I would recommend toning-down the speculative claims in the abstract as they are not representative of the article as a whole.

We have presented our data, performed the statistical analysis, and are presenting conclusions based on statistically significant findings.

It would be useful in the introduction to briefly define age-related hearing loss. 

We have added the following sentence in our introduction, line 46-47:

"Age-related hearing loss refers to a degenerative process of aging resulting in a progressive bilateral hearing loss. (Nelson EG, Hinojosa R. Laryngoscope 2006;116 (Suppl 112):1-12.)"

What sort of thresholds are considered significant? Who’s definition is this? You can then refer to this in the methods section when defining loss in the context of the study.

Our classification 

---

## [Decision Letter · Decision Letter 1]

6 Nov 2024

PONE-D-24-15625R1Measurement of inferior colliculus volume  based on MRI image stacks and its relationship with age and hearing statusPLOS ONE

Dear Dr. Kwok,

Thank you for submitting your manuscript to PLOS ONE. After careful consideration, we feel that it has merit but does not fully meet PLOS ONE’s publication criteria as it currently stands. Therefore, we invite you to submit a revised version of the manuscript that addresses the points raised during the review process.

Thank you for your effort in improving the manuscript. However, the reviewer believes the author has not responded adequately. Please revise the points raised in this and previous comments, especially those that have not been sufficiently addressed. If you disagree with the reviewer's comments, please politely rebut them. If you disagree with the reviewer's comments, please rebut them politely. If the same happens again, I might consider the review process unsuccessful and reject the manuscript.

We look forward to receiving your revised manuscript.

Kind regards,

Ryota Sakurai, Ph.D.

Academic Editor

PLOS ONE

**Additional Editor Comments:**

Thank you for your effort in improving the manuscript. However, the reviewer believes the author has not responded adequately. Please revise the points raised in this and previous comments, especially those that have not been sufficiently addressed. If you disagree with the reviewer's comments, please politely rebut them. If you disagree with the reviewer's comments, please rebut them politely. If the same happens again, I might consider the review process unsuccessful and reject the manuscript.

Reviewers' comments:

Reviewer's Responses to Questions

**Comments to the Author**

1. If the authors have adequately addressed your comments raised in a previous round of review and you feel that this manuscript is now acceptable for publication, you may indicate that here to bypass the “Comments to the Author” section, enter your conflict of interest statement in the “Confidential to Editor” section, and submit your "Accept" recommendation.

Reviewer #1: All comments have been addressed

Reviewer #2: (No Response)

2. Is the manuscript technically sound, and do the data support the conclusions?

Reviewer #1: Partly

Reviewer #2: Yes

3. Has the statistical analysis been performed appropriately and rigorously? 

Reviewer #1: Yes

Reviewer #2: Yes

4. Have the authors made all data underlying the findings in their manuscript fully available?

Reviewer #1: Yes

Reviewer #2: Yes

5. Is the manuscript presented in an intelligible fashion and written in standard English?

Reviewer #1: Yes

Reviewer #2: Yes

6. Review Comments to the Author

Reviewer #1: Dear editor

Initially, I would like to express my displeasure at receiving a review without indications of modifications in the body of the text.

Another point, the indication of the use of the classification of the degree of hearing loss is still not mentioned in the text. It's no use mentioning that you used the reference and not presenting this data to the reader.

Regarding the use of the 5kHz stimulus, it is well known that the Biologic equipment performs the analysis at this frequency in all its versions. And it is important to say that there is no recommendation for its use at 5kHz and there never has been. Concurrently, this frequency is not analyzed in the PTA. Therefore, the responses are not suitable for publication.

The responses are vague and the insertions made did not significantly improve the quality of the manuscript. Therefore, I do not recommend the publication of the manuscript.

Reviewer #2: Thank you for your diligent attention to the reviewers' comments. I feel you have significantly improved the paper and that it is nearly ready for publication.

I have one further comment and suggestion to make on the subject of describing in your methods section how the edge of the IC was determined. I believe it is the responsibility of authors to describe all methods used such that they can be replicated by a reader. I do not believe this is "trivial" or that it "goes beyond the scope of the description in the methods".

Please *briefly* describe the process used for drawing the ROI so that a reader could try and follow the process, or cite an article that does so.

7. PLOS authors have the option to publish the peer review history of their article (what does this mean?). If published, this will include your full peer review and any attached files.

Reviewer #1: No

Reviewer #2: **Yes: **Rebecca Susan Dewey

---

## [Author Response · Author response to Decision Letter 1]

10 Dec 2024

Detailed response to the reviewers´ comments:

Reviewer #1

Initially, I would like to express my displeasure at receiving a review without indications of modifications in the body of the text.

As required for the revision, we submitted a marked-up copy of the manuscript named “revManuscript with tracking.docx” that highlights all the changes made to the original manuscript version. The modifications were indicated by the “markup function” of Microsoft Word. We have no explanation why the reviewer could not see the marked modifications made during the revision. 

Another point, the indication of the use of the classification of the degree of hearing loss is still not mentioned in the text. It's no use mentioning that you used the reference and not presenting this data to the reader.

We now outline the definition of the different grades of hearing loss that were defined in the “Report of the Informal Working Group on Prevention of Deafness and Hearing Impairment Programme Planning, Geneva, 18-21 June 1991” in the methods and include this WHO report it in the references as number [29] “World Health Organization. (1991). Report of the informal working group on prevention of deafness and hearing impairment programme planning: Geneva, 18–21 June 1991. Geneva: World Health Organization. Retrieved from https://apps.who.int/iris/handle/10665/58839.” 

Our classification is based on the grades defined in the report of the World Health Organization. (1991). To reduce the number of groups and increase the number of samples within each group for statistical analysis, we defined 3 categories by combining the grades 0 (no impairment, <25dB) and 1 (slight impairment, 26-40dB) as category 1 (≤40dB), grades 2 (moderate impairment, 41-60dB) and 3 (severe impairment, 61-80dB) as category 2 (41-80dB) and grade 4 (profound impairment including deafness, >81dB) as category 3 (see now end of method section). 

Regarding the use of the 5kHz stimulus, it is well known that the Biologic equipment performs the analysis at this frequency in all its versions. And it is important to say that there is no recommendation for its use at 5kHz and there never has been. Concurrently, this frequency is not analyzed in the PTA. Therefore, the responses are not suitable for publication.

As explained in the initial revision, this study is based on a retrospective analysis of cases from the ENT department of the University of Regensburg. BERA is routinely used in our clinic to diagnose hearing status at 3 frequencies in babies where PTA is not possible. Before 2017 the high frequency tested was 5 kHz which was changed to 4 kHz after 2017. 

We disagree with the reviewer´s notion that the determination of the hearing status by BERA can only be evaluated with a test frequency of 4 kHz but not with 5 kHz. Assuming a 35mm long basilar membrane, the position of the 4 kHz place from the cochlear base is ≈11.6 mm or 33% while that of the 5 kHz place is ≈10.0 mm or 29% from the base (according to Greenwood, 1961). Thus, based on the logarithmic frequency representation along the human cochlea, the difference between these two frequencies is not dramatic. In addition the hearing class is based on the average threshold of the 3 BERA frequencies tested. We do not see that the BERA determination of threshold at 5 kHz versus 4 kHz would substantially affect the outcome of the hearing loss classification. 

The responses are vague and the insertions made did not significantly improve the quality of the manuscript. Therefore, I do not recommend the publication of the manuscript.

In the first revision we comprehensively answered and discussed 16 points raised by the reviewer and made corresponding changes and additions to the manuscript. 

The points initially raised by the reviewer and our responses are repeated below. We regard these responses as specific and detailed and believe that the corresponding changes to the manuscript improved the manuscript. We disagree with the global classification of our responses to the 16 points raised in the first review by this reviewer as “vague”.

“You mentioned that all patients have PTA or BERA. However, it is crucial to specify the type of BERA that was employed, as the BERA-click does not provide data that is comparable to PTA. Therefore, examine this item and substitute the procedure name for BERA-specific frequency.

For the present exploratory retrospective analysis of the relation of hearing status and IC volume we noticed early during the study the heterogeneity of the subjects with regard to age and hearing status (e.g. Fig. 4). Thus, we focused on a coarse classification of hearing loss that can be achieved by evaluating standard clinical PTAs and BERAS performed in the audiology section of the Regensburg ENT clinic. 

To obtain frequency specific information we recorded in Regensburg the response to short tone bursts which were used to evaluate the hearing status. For measuring the interpeak-latences I-V click stimuli were used. 

We have added a sentence concerning this issue in lines 137-139 of the manuscript.

PTA – indicating the values and equipment that are used.

See Raw data (S1 Table) that are now provided in “supplementary data” and give the thresholds obtained at the PTA test-frequencies and for the BERA stimuli.

The equipment used for PTA: Madsen Astera, Co. Otometrics/Münster-Germany

The equipment used for BERA: Navigator Pro, Bio-Logic/Natus, Model 580-NVBOX1156A, Worthing, UK.

The BERA frequency specifies the values and equipment that are employed. Before the specified frequency was reached, BERA was executed–click?

To obtain frequency specific information we recorded in Regensburg the response to short tone bursts which were used to evaluate the hearing status. For measuring the interpeak-latences I-V click stimuli were used. 

Why was 5kHz employed? This frequency is not of great significance within an audio battery.

For the BERA values in this study, the Navigator Pro-machine was used which had been preset for the examination with click-stimuli using the upper frequency limit at 5kHz. 

In our newer BERA equipment (Eclipse by Interacoustics, Middelfart, Denmark) the upper frequency limit for clicks is now 4kHz.

However, as this is a retrospective study, the examination frequency cannot be changed to 4kHz. Consequently, high frequency loss in the BERA measurements was determined in the majority of earlier data by 5kHz and in more recent measurements with a 4kHz setting. 

It is strongly recommended that auditory loss be classified in accordance with the guidelines of the World Health Organization (2019). And that the categories be divided by this classification.

Our classification is in close accordance with the classification mentioned above. It combines the grades 0 (no impairment, <25dB) and 1 (slight impairment, 26-40dB) as category 1 (normal to mild hearing loss) and grade 4 (profound impairment including deafness, >81dB) as category 3 (severe hearing loss or deafness) as listed in lines 148-154. 

I would have appreciated the opportunity to review a prior study that examined the number of participants who would be considered reliable for this study. 

Since this exploratory retrospective study was, to our knowledge, the first systematic trying to correlate the relation of IC-volume with other factors like age or hearing status, these data can provide the basis to estimate the required sample size for more detailed future studies. 

Note that the size of the letters is greater than the remainder of the manuscript on page 10, lines 182 to 184.

This formatting mistake has been corrected. Due to corrections in the manuscript the lines have shifted to 189-191.

It is essential to mention in a specific section of the manuscript that the PTA is the standard for diagnosing auditory loss in children who are capable of responding appropriately to the elderly.

PTA was used in older children and adults when possible. BERA was used in babies when a PTA could not be reliably measured. In some deaf children over 1 year a BERA was performed in the course of Cochlear Implant diagnostics. 

This we now mentioned in the methods part lines 132-137:

The PTAs are the standard for diagnosing auditory loss in children who are capable of responding appropriately to the examiner. PTA was used in 67 children and adults and determined air-conduction thresholds using 9 frequencies (125Hz to 8kHz). The BERA was used in 56 young children (aged 3 months to 4 years) in whom a PTA could not be reliably measured, and at least 3 frequencies were included (500-1000Hz, 2kHz, 5kHz). In some deaf children over 1 year of age, a BERA had been performed in the course of cochlear implant diagnostics.

In the discussion, the first sentence indicates that hearing loss is more prevalent after the age of 70. I suggest reading additional articles, as there is already evidence that it occurs more frequently today. 

The National Institute on Deafness and Other Communication Disorders states in their Quick Statistics about Hearing, Balance & Dizziness Report which was updated Sept. 20, 2024, that among adults ages 20-69 who report 5 or more years of exposure to very loud noise at work, about 18% have speech-frequency hearing loss in both ears. About 5% of adults ages 45-54 have disabling hearing loss. The rate increases to 10% for adults ages 55-64. 22% of those ages 65-74 and 55% of those who are 75 and older have disabling hearing loss.

These numbers do not differ so much from the ones stated in our discussion. 

Another study from 2010 looked at hearing thresholds from 1999-2004 and compared them with thresholds from 1959-1962 ( Hoffman HJ et al. Americans hear as well or better today compared with 40 years ago: hearing threshold levels in the unscreened adult population of the United States, 1959-1962 and 1999-2004. Ear Hear 2010 Dec;31(6):725-34.) They found that prevalence of overall hearing impairment was significantly lower in 1999-2004 than in 1959-1962.

However, the population worldwide is becoming older and, therefore, age-related hearing loss can be found more frequently than in the past. 

Additionally, I suggest that the discussion regarding prevalence and incidence be more in-depth, including, for instance, the fact that adolescents and children tend to lose their hearing even more rapidly as a result of the use of hearing aids. The initial paragraph is extremely rudimentary and inadequate. 

The aim of this study was to examine the volumes of the inferior colliculus and relate them with age and hearing loss. A discussion regarding prevalence of hearing loss and the impact of the use of hearing aids on hearing in adolescents would be suitable for a review study but not for this retrospective analysis, which cannot find answers for the changes in prevalence of hearing loss.

In terms of the IC volume, it was determined that two individuals exhibited values that were higher. I suggest that they draw conclusions about this. What was the profession of these individuals? Were they musicians? In conclusion, they must better package this item. 

This is an interesting point and we looked up these two cases.

Both patients with the high IC volumes happened to be deaf children (3 months and 4 years old), so they were certainly not musicians. We could not draw any sound conclusions from the results.

Furthermore, when discussing the IC volume in relation to left/right, they also mentioned left/right-handedness very appropriately. However, did you analyze this variable? 

Unfortunately, we have no records of the handedness. 

It would be an extremely significant point in this study. Given the impossibility of this analysis, as it is a retrospective study, I suggest that it be included as a constraint on the study. 

The following sentence was added to the discussion lines 387-388

It is a constraint of this study, that handedness cannot not be considered in the discussion of the IC volume in relation to right/left sides (lateralization of hearing), as there were no records available.

I suggest that the study's limiting factors, which encompass a multitude of other factors, be updated, and that new studies be proposed. 

This sentence was added at the end of the discussion line 416-418:

Since this exploratory retrospective study was, to our knowledge, the first systematic trying to correlate the relation of IC-volume with other factors like age or hearing status, these data can provide the basis for focused study designs comparing age matched groups with well-defined hearing status and for the estimation of the required sample size for more detailed future studies. 

Furthermore, it is not possible to generalize and assert that there is no symmetry, which would imply that this was observed in the current study with a reduced number of participants. 

A Wilcoxon paired sample test comparing left and right IC volume of the 123 subjects found no systematic significant difference or systematic asymmetry between both sides (p = 0.865). 

We do not state in the manuscript that “there is no symmetry”, rather we describe that the data “show no systematic difference … between both sides.” and this is also reflected in the discussion lines 381 – 386. 

The same applies to the other apprehensions. This study must be replicated with an appropriate number of participants, as well as more clearly defined evaluation procedures.

We agree that our retrospective explorative analysis cannot unequivocally answer many detailed questions. But it provides the basis for the focused design of future studies with adequate estimates of required sample size.” 

As illustrated by the answers above, we do not consider our responses to the questions raised by the reviewer in the first revision as “vague”. 

Reviewer #2: 

I have one further comment and suggestion to make on the subject of describing in your methods section how the edge of the IC was determined. I believe it is the responsibility of authors to describe all methods used such that they can be replicated by a reader. I do not believe this is "trivial" or that it "goes beyond the scope of the description in the methods".

Please *briefly* describe the process used for drawing the ROI so that a reader could try and follow the process, or cite an article that does so.

The corpora quadrigemina consist of the inferior (more caudal) and superior (more rostral) colliculi and are elevations of the tectum forming the dorsal surface of the midbrain. As illustrated in the original orthogonal MRI views in Fig. 1A, the surface of the IC is well defined and clearly recognizable. The body of the IC below its surface appears as brighter area of the tectum. Fig. 1B illustrates how voxels within these borders of the IC were manually labelled as "Region of Interest" to determine the number of voxels belonging to the left and right IC (Fig.1A and Fig.1B). 

This description has been added to the method section.

---

## [Decision Letter · Decision Letter 2]

27 Dec 2024

Measurement of inferior colliculus volume  based on MRI image stacks and its relationship with age and hearing status

PONE-D-24-15625R2

Dear Dr. Kwok,

We’re pleased to inform you that your manuscript has been judged scientifically suitable for publication and will be formally accepted for publication once it meets all outstanding technical requirements.

Kind regards,

Ryota Sakurai, Ph.D.

Academic Editor

PLOS ONE

Additional Editor Comments (optional):

The manuscript has been improved following the editor and reviewers suggestions. Thanks for your effort.

Reviewers' comments:

Reviewer's Responses to Questions

**Comments to the Author**

1. If the authors have adequately addressed your comments raised in a previous round of review and you feel that this manuscript is now acceptable for publication, you may indicate that here to bypass the “Comments to the Author” section, enter your conflict of interest statement in the “Confidential to Editor” section, and submit your "Accept" recommendation.

Reviewer #2: All comments have been addressed

2. Is the manuscript technically sound, and do the data support the conclusions?

Reviewer #2: Yes

3. Has the statistical analysis been performed appropriately and rigorously? 

Reviewer #2: Yes

4. Have the authors made all data underlying the findings in their manuscript fully available?

Reviewer #2: Yes

5. Is the manuscript presented in an intelligible fashion and written in standard English?

Reviewer #2: Yes

6. Review Comments to the Author

Reviewer #2: I feel my comments have been adequately addressed. However, I see that Reviewer 1's comments have not been addressed to their satisfaction. This is disappointing and I recommend that the Authors address these shortcomings before further attempting publication.

7. PLOS authors have the option to publish the peer review history of their article (what does this mean?). If published, this will include your full peer review and any attached files.

Reviewer #2: **Yes: **Rebecca Susan Dewey

---

## [Editor Report · Acceptance letter]

17 Jan 2025

PONE-D-24-15625R2 

PLOS ONE

Dear Dr. Kwok, 

I'm pleased to inform you that your manuscript has been deemed suitable for publication in PLOS ONE. Congratulations! Your manuscript is now being handed over to our production team.

Kind regards, 

on behalf of

Dr. Ryota Sakurai 

Academic Editor

PLOS ONE